# Spatially Resolved Correlation between Stiffness Increase and Actin Aggregation around Nanofibers Internalized in Living Macrophages

**DOI:** 10.3390/ma13143235

**Published:** 2020-07-21

**Authors:** Guoqiao Zhou, Bokai Zhang, Liyu Wei, Han Zhang, Massimiliano Galluzzi, Jiangyu Li

**Affiliations:** 1Shenzhen Key Laboratory of Nanobiomechanics, Shenzhen Institutes of Advanced Technology, Chinese Academy of Sciences, Shenzhen 518055, China; gq.zhou@siat.ac.cn (G.Z.); bk.zhang@siat.ac.cn (B.Z.); ly.wei@siat.ac.cn (L.W.); zhanghan2010@my.swjtu.edu.cn (H.Z.); 2University of Chinese Academy of Sciences, Beijing 100049, China; 3State Key Laboratory of Traction Power, Southwest Jiaotong University, Chengdu 610000, China

**Keywords:** atomic force microscopy (AFM), living cells mechanics, finite element simulations, hydrogel nanofibers

## Abstract

Plasticity and functional diversity of macrophages play an important role in resisting pathogens invasion, tumor progression and tissue repair. At present, nanodrug formulations are becoming increasingly important to induce and control the functional diversity of macrophages. In this framework, the internalization process of nanodrugs is co-regulated by a complex interplay of biochemistry, cell physiology and cell mechanics. From a biophysical perspective, little is known about cellular mechanics’ modulation induced by the nanodrug carrier’s internalization. In this study, we used the polylactic-co-glycolic acid (PLGA)–polyethylene glycol (PEG) nanofibers as a model drug carrier, and we investigated their influence on macrophage mechanics. Interestingly, the nanofibers internalized in macrophages induced a local increase of stiffness detected by atomic force microscopy (AFM) nanomechanical investigation. Confocal laser scanning microscopy revealed a thickening of actin filaments around nanofibers during the internalization process. Following geometry and mechanical properties by AFM, indentation experiments are virtualized in a finite element model simulation. It turned out that it is necessary to include an additional actin wrapping layer around nanofiber in order to achieve similar reaction force of AFM experiments, consistent with confocal observation. The quantitative investigation of actin reconfiguration around internalized nanofibers can be exploited to develop novel strategies for drug delivery.

## 1. Introduction

Macrophages are the most plastic cells in the hematopoietic system, existing in all tissues with great diversity [1]. They play a protective role in pathogen immunity, tissue development, homeostasis and regeneration after injury [2], responding to signal fluctuation of microenvironment by changing their function rapidly. Drug treatment is one of the most important approaches to achieve macrophage functionalization [3], and in addition to the widely used chemical drugs, nanomedicines are attractive due to their controlled drug release, prolonged half-life and lower side effects [4]. Indeed, nanoparticles have been used in therapeutic applications of anti-inflammation and macrophage-related disorders [5] and to inhibit tumor growth by inducing pro-inflammatory macrophage polarization [6]. In some cases, however, macrophages mutate into a pro-healing phenotype, resulting in tumor-associated macrophages (TAMs) that promote the metastasis, shifting tumors towards an immunosuppressive microenvironment, and thus increasing the resistance to cancer therapy [7]. On the other hand, nanoparticles of immunotherapy for cancer treatment can induce endothelial leakiness [8]. In such a situation, both the blood stream circulating macrophages and the tumor accompanying TAMs tend to penetrate the vessel wall and compete with each other regarding tumor growth. Therefore, the effects of drug treatment on mechanical properties of macrophages are pivotal to understand the potential benefit and drawbacks of nanomedicines.

Considering macrophages as drug targets, recent studies have demonstrated correlation between their mechanical properties and drug treatment [9,10]. For example, when using lipopolysaccharide (LPS) to trigger the M1 phenotype differentiation, an increase in the elastic modulus of macrophage was observed [11,12]. Furthermore, internalized silver nanoparticles caused cytoskeleton stiffening upon low to medium uptakes, while high uptake disintegrated the actin network, resulting in cytotoxicity [13]. However, it was also reported that macrophages interacting with ZnO nanoparticles showed a decrease in cytoskeleton stiffness [14,15]. In general, nanoparticles’ cytotoxicity is initiated by increased oxidative stresses, followed by protein redox modifications such as actin filaments disassembly and inflammatory cascade leading to cell death pathways [16]. Nevertheless, the restructuration of cytoskeleton triggered by nanoparticles below toxicity level is not well understand, especially the microcellular mechanisms that are responsible for the change in cell stiffness. In order to study this problem, spatially resolved imaging of stiffness and microcellular structure in living macrophages is necessary, especially in a quantitative manner with high spatial resolution.

To this end, we use biocompatible polylactic-co-glycolic acid (PLGA)–polyethylene glycol (PEG) hydrogel [17] nanofibers as a model drug carrier system mimicking the shape and size of bacteria, which were reported to be internalized with a high rate, ideal for drug-delivery [18]. We study the restructuration of actin in RAW 264.7 macrophages upon internalization, since this cell line is particularly suited to model the response of macrophages in higher organisms during clearance of foreign bodies and apoptotic debris [1]. Atomic force microscopy (AFM) is a powerful tool to detect variation of mechanical properties of actin correlated to pathophysiological conditions [19,20,21,22], microenvironment [23] and drug treatment focused on cytoskeleton [24,25,26], and therefore, we use AFM integrated with fluorescence microscope to simultaneously map the cell Young’s modulus and morphology in correspondence to the localized PLGA-PEG nanofibers, and observe strong correlation. Finite element simulation (FEM) is also applied to reconcile the effect of actin aggregation around nanofibers, as intake of nanofibers alone cannot explain the observed local Young’s modulus increase. The effect is further supported by confocal laser scanning microscopy (CLSM).

## 2. Materials and Methods

### 2.1. Nanofibers Preparation

The nanofibers used in this study were prepared via a two-step process following the procedure reported by Zhang et al. [27]. Firstly, the PLGA-PEG bulk polymers (PLGA_57500_-PEG_5000_) were transformed into spherical shaped particles via the precipitation/solvent diffusion method [28]. Briefly, 50 mg PLGA-PEG and 0.5 mg Nile Red (Invitrogen, Waltham, MA, USA) were dissolved in 1.25 mL of dichloromethane (DCM) (Sigma-Aldrich, St. Louis, MO, USA) for at least 3 h. The polymer solution was directly added to 5 mL of 5% Poly(vinyl alcohol) (PVA) (Mw 31,000–50,000 Da, 87–89% hydrolyzed) solution. The mixture was then homogenized for 1 min by using a probe sonicator at 9 W to generate an oil-in-water (O/W) emulsion. The formed emulsion was added to 25 mL of ice-cold Deionized (DI) water and stirred for 3 h at room temperature to evaporate DCM. After that, the particles were collected and washed by centrifugation at 20,000 g force at 4 °C, three times, and then passed through a 1.2 μm filter, resulting in ~450 nm diameter spherical shaped PLGA-PEG particles. The particles were freeze-dried and stored at −20 °C for later use. Secondly, the as-synthesized spherical shaped PLGA-PEG particles were stretched into needle shape via the reported stretching method [29]. Briefly, spherical shaped PLGA-PEG particles were added to a solution with 10% PVA, and mixed well by magnetic stirring. 15 mL of the solution was dried on a 12 × 16 cm^2^ flat surface to form an 80 μm thickness film. The film was cut into sections (1 × 16 cm), and stretched along the longitudinal direction at a temperature higher than 70 °C in air by using the hair drier, until their length reached 64 cm. The finished films were dissolved in ice-cold DI water to release the embedded nanofibers. The nanofibers were then washed by centrifugation at 20,000 g force with DI water at 4 °C, at least 5 times, to remove the residual PVA. The nanofibers were finally freeze-dried, weighted and stored at −20 °C for further use.

### 2.2. Living Cells Culture Protocol

RAW 264.7 (ATCC^®^ TIB-71TM) murine macrophage cells were used in this study. The cells were cultured with complete culture media containing Dulbecco’s modified Eagle’s media (Gibco, Grand Island, NY, USA), supplemented with 10% fetal bovine serum (FBS; Sigma-Aldrich, St. Louis, MO, USA), 100 units/mL penicillin and 100 units/mL streptomycin. Cells were grown in a standard cell culture incubator at 37 °C with 5% CO_2_ in a humidified atmosphere. Cells were allowed to be incubated for 24 h before PLGA-PEG fibers were introduced for both CLSM and AFM studies. For AFM experiments, the cultured cells were detached from the flask by scraping, and seeded in culture media without FBS in a 35 mm diameter, 1 mm thick culture dish with the round glass slide for AFM. After 24 h, the media was discarded, and replaced by PLGA-PEG nanofibers (100 μg/mL) in culture media for another 14 h incubation. Finally, the glass slide with cells was washed with culture media twice and placed in a BioHeater (from Asylum Research, Santa Barbara, CA, USA) with culture media for the AFM measurement. In order to thermalize the culture plates at 37 °C, the environmental controller with BioHeater was used during all AFM experiments.

### 2.3. Measuring the Young’s Modulus of Living Cells

The complete protocol and methodology for nanomechanical measurements on soft matter, specifically living cells, is described in a series of methodological studies from our group [30,31,32]. Briefly, topographic and mechanical imaging was performed using a MFP3D-Bio AFM from Asylum Research in Force Mapping mode, resulting in a force vs. indentation curve (force curve, FC) for each point of a regular square grid on the surface [33]. We used spherical colloidal probes (Novascan, Milwaukee, WI, USA) made of borosilicate glass, having a typical nominal spring constant k = 0.2 N/m and radius R = 5 μm. The radius of the sphere was characterized by means of AFM reverse imaging of the colloidal probe on a spiked grating TGT1 (NT-MDT, Moscow, Russia) [34]. Spring constant and optical lever sensitivity were calibrated via force curves on a hard surface (glass) and an integrated thermal noise routine from Asylum Research. We selected the standard parameters for the acquisition of FC as follows: ramp size 8 µm, force setpoint F_MAX_ ≈ 5–7 nN (0.5 V deflection, converted after calibration in force), approaching velocity v = 32 µm/s, ramp rate 2 Hz and sampling rate 10 KHz. A total of 32 × 32 = 1024 force curves were typically acquired in each force mapping at medium resolution in a scan time of 9 min, allowing the acquisition of a population of 20–30 cells, while 1–2 cells were acquired at high-resolution (64 × 64). The raw (compressed) topographic maps of the cells were built using the local z-position corresponding to the maximum setpoint force. All the experiments were performed in phosphate buffer saline (PBS) solutions at constant temperature of 37 °C, maintained using BioHeater stage from Asylum Research.

### 2.4. Laser Scanning Confocal Microscopy

Confocal laser scanning microscopy (CLSM, Leica SP5TCS II, Wetzlar, Germany) was used to study the cellular uptake of the PLGA-PEG nanofibers as well as the actin distribution during the uptake process. Briefly, cells were firstly incubated with medium containing 100 μg/mL nanofibers for 14 h in a confocal dish. Cells were then washed with PBS, fixed with 4% formaldehyde, permeabilized with 0.1% Triton X-100 and stained with Phalloidin (Invitrogen). After 30 min incubation at 25 °C, cells were washed and dried, and mounted with VECTASHIELD Antifade Mounting Medium containing 4′, 6-diamidino-2-phenylin-dole (DAPI) for nuclei. Fluorescent signals were collected using excitation λ = 405 nm and emission λ = 461 nm (blue) for DAPI, while excitation λ = 488 nm and emission λ = 516 nm (green) for Phalloidin.

### 2.5. Finite Element Simulations

A three-dimensional (3D) finite element (FE) model was developed using the preprocessing software HyperMesh to study indentation on living cell systems with a spherical indenter, mimicking AFM experiments. Protrusions of the living cell are in irregular shapes, therefore, for simplicity, we use a combination of spherical crown and semi-ellipsoid to mimic the cell body joined with protrusion. The resulting model requires several strategies to reduce computation and improve precision: (i) according to symmetry, only half of the model was established, (ii) a finer mesh of the nanofiber and interphase covering the nanofiber were created to improve precision and (iii) the transition region in the vicinity of interphase was intentionally meshed in a biased way to connect the rough regions that are mostly insensitive to the external load. This interphase region around the nanofiber is also defined as a wrap layer, in order to describe the dense network of actin, as described later in the Results and Discussion Section. The overall size of the entire specimen is taken as 30 μm in lateral span (radius) and 6 μm in vertical span (height). The protrusion length is selected as 15 μm according to the optical and AFM investigations. The radius of the spherical indenter is set as 5 μm. The bottom boundary is fixed to substrate and no movement is allowed. The cell surface is generally free without constraint, except after establishing contact with the probe, where the cell membrane is forced to adhere and follow probe geometry.

The finite element simulation was performed on ABAQUS. The cell body, nanofibers and interphase are represented by neo-Hookean hyperelastic material [35], having hyperelastic parameter *C*_10_ = *E*/6, while *E* is the initial Young’s modulus for shallow indentation. The Neo-Hookean model was chosen in order to improve the computational efficiency. Moreover, the non-linearity introduced by this model is small in comparison with linear Hertzian mechanics [36]. Therefore, we built the hyperelastic parameter *C*_10_ based on AFM experimental data of a series of shallow indentation on the cell body, which was estimated to have initial *E*_cell_ = 150 Pa, and Poisson’s ratio ν_cell_ = 0.5. The geometry and mechanical properties of nanofibers and interphase, both having Poisson’s ratio of 0.5, were varied to produce a set of simulations to be compared with AFM data. Finally, the Young’s modulus and Poisson’s ratio of the probe were set to be 160 GPa and 0.22, respectively. 

We exerted external load as in AFM by displacing the spherical indenter in a direction normally to the substrate until maximum indentation of 1 μm. For composite simulations, such as cell + nanofiber + wrap, the calculation is diverging and stops at 600 nm indentation, equivalent to 3.7 nN of load force. Therefore, in the Results and Discussion Section, we represent force vs. indentation curves from FEM calculations up to 3.7 nN, while in the Appendix A, we used 600 nm constant indentation.

## 3. Results

We started our study with the usage of AFM integrated with fluorescence microscopy, which had the capability to image the cell morphology and microcellular structure while simultaneously acquiring the force curve of AFM indentation (Figure 1) [37,38].

This enabled us to correlate the structure change and quantitative nanomechanical variation induced by the nanofiber internalization. PLGA-PEG nanofibers embedded with Nile Red were fed to RAW 264.7 macrophages. Transmission optical microscopy (Figure 2a) clearly showed the cell morphology after internalization, while the fluorescence microscopy (Figure 2b) revealed nanofibers as marked by the white ovals, most of them located at the edge and the protrusion of the cells.

The cell morphology is also shown by AFM topography (Figure 2c) in good agreement with the optical image, and the corresponding mapping of Young’s modulus after 14 h of internalization is presented in Figure 2d, acquired from a 64 × 64 grid of force curves (force volume) using a spherical probe with a 5 μm radius. For the analysis of force volume, the contact part of each force curve was individuated by binning the force axis and producing a histogram, the non-contact part was determined as a sharply defined Gaussian distribution, peaked at zero force. The region of the force curve above the width of the Gaussian distribution was considered as the indentation for the fitting procedure. The contact part was then analyzed using the modified Hertz model for finite thickness samples (Equation (1)) [39]:(1)$$F=\frac{4}{3}\frac{E\sqrt{R}}{\left(1-\nu^{2}\right)}\delta^{3/2}\left[1+1.009\chi_{S}+1.032\chi_{S}{}^{2}+0.578\chi_{S}{}^{3}+0.0048\chi_{S}{}^{4}\right]$$
where *F* is the applied force, *δ* is the indentation, *ν* is the Poisson’s ratio, *E* is the effective Young’s modulus of the cell, *R* is the radius of the spherical probe and $\chi_{S}=\sqrt{R\delta }/h$ is a dimensional parameter. The fitting was performed using an indentation interval from 0% to 100% of maximum indentation. The correction from Dimitriadis et al. [39] was designed for two extreme boundary conditions: bound layer (i.e., well adherent) and free-to-move layer. Since cells are alive, they can dynamically move and adhere on substrate using focal adhesions points, therefore we use a boundary condition between the bound and not-bound limits, by using the arithmetic mean of coefficient for bound and not-bound states [32].

Combining all four images of Figure 2, we noticed that large cell protrusions containing nanofibers, as highlighted by ovals, resulted in a considerable increase in Young’s Modulus from 150–200 Pa of the cell body to 1000–2000 Pa around nanofibers.

Interestingly, there are also other regions showing an increase in Young’s modulus, which can be ascribed to the influence of some out of focus nanofibers as well as organelles, especially the nucleus [40].

Having established the correlation between the localized Young’s Modulus increase around nanofibers, we studied the effect in more depth, comparing control (Figure 3a–d) and cells internalized with nanofibers (Figure 3e–h).

Both cells showed morphology constituted by a central body with several external protrusions (Figure 3a,e), suggesting that immune polarization was not induced [41]. Control exhibited Young’s Modulus of 161 ± 12 Pa, as revealed by mapping and the histogram in Figure 3b,c. This was much softer than other cell lines (different from macrophages), reported to be around 500–2000 Pa and acquired using the same technique [42]. Such low Young’s Modulus may reflect the structural organization of actin in macrophages for higher deformability, especially during internalization of large bodies [43]. RAW264.7 internalized with nanofibers, on the other hand, exhibited much larger Young’s modulus of 251 ± 18 Pa, as revealed by mapping and the histogram in Figure 3f,g. The correlation between the internalized nanofiber and the enhanced Young’s modulus is shown in Appendix A in the Appendix A, acquired via simultaneous AFM and fluorescence microscopy. The contrast was also evidenced in the comparison of single force curves in Figure 3d,h, selected on cell areas representing normal soft cell body (circles) and hard regions (triangles) representing nucleus in the control [40] and nanofiber inside the cell. 

For each cell population condition, the mean Young’s modulus value, *E*, and its associated effective errors (standard deviation of the mean, *σ_E_*) have been calculated, following the procedure described in details in References [31,32]. Briefly, for a single *FV* measurement, representing a single cell out of a total of *N* cells studied at a given condition, the error *σ_FV_* associated with the mean Young’s modulus of the cell was calculated by taking into account the propagated uncertainty from the probe calibration, and the variability of Young’s modulus within the cell as the width of logarithmic distribution. At this point, the center, *µ*_10_, and standard deviation, *σ*_10_, from semilog10 scale were converted in linear values using the linearization formula from lognormal distributions, (Equation (2)), i.e.,:
(2)$$\begin{array}{cc} E_{FV}=10^{\mu_{10}+\left(0.5\;ln\;10\right)\sigma_{10}^{2}}\;\;\;\;\;\;\; & \;\;\;\;\;\;\;\sigma_{FV}=E_{FV}\sqrt{10^{\sigma_{10}^{2}}-1} \end{array}$$

Then, the average Young’s modulus, *E,* of cells in a given condition was calculated as the mean of *N* single cell mean values (the population mean), with its standard deviation of the mean *σ*_mean_. Eventually, the error *σ_E_* associated with *E* (an effective standard deviation of the mean) was calculated as the sum in quadrature of *σ*_mean_ and the average *σ_FV_*. The results are therefore presented as *E* ± *σ_E_*. The statistical significance of the differences of Young’s modulus values of different conditions are calculated using the two-tailed Student’s t-test and considering *p*-value.

Finally, the average Young’s Modulus was statistically evaluated on a population of 20–30 single cells (Table 1), showing a significant increase of Young’s modulus of cells due to the internalization of nanofibers from 148.8 ± 4.4 to 215.9 ± 7.9 Pa, with *p*-value < 0.0001 when comparing to control.

Notice that we used finite thickness correction on a standard Hertz model for spherical indentation [39], which is important to analyze all the cell parts avoiding the influence of substrate.

The observed stiffness increase around nanofiber was expected, since PLGA-PEG with Young’s modulus around 1 MPa is much stiffer, as shown in Appendix A. But, can nanofiber alone explain the increased stiffness, or is there microcellular structure reconstruction involved? To answer this question, we examined RAW264.7 internalized with PLGA-PEG nanofibers using confocal laser scanning microscopy (CLSM). After internalization, phalloidin was used to stain actin filaments [44]. Note that actin distribution in control cells was homogenous, without local accumulations, as shown in Appendix A. The red color from nanofibers in the projected CLSM images along 3D directions (Figure 4a) confirms their internalization, as the signal distribution was inside the cell.

During internalization, the most obvious characteristic was the actin-driven protrusion always covering the internalized target [29,45]. Such evidence in our case came from the yellow color highlighted by a white arrow in Figure 4a, obtained by overlapping green and red signals, indicating that nanofibers can induce a redistribution of actin to internalize the nanofibers located in an external cell protrusion. It is worth noticing that the intense fluorescence signal in Appendix A from the central part of the cell and near the nucleus has no correspondence in the Young’s modulus map in Appendix A; in fact, actin is mostly recruited within the protrusion during early phagosomes formation, while the actin network is dismantled for late phagosomes transported inside cytosol near the nucleus [46,47]. This behavior is shown in Figure 4a and in additional confocal images in Appendix A. Finally, to understand the effect of this actin aggregation on cell nanomechanics, we resorted to FEM simulations. 

A 3D FEM model was created to simulate indentation by AFM localized on top of an internalized nanofiber, as shown in Figure 4b. The geometrical parameters of the cell were derived from AFM morphology, while the Young’s modulus of the cell body, nanofibers and actin wrap used in the simulation are 150 Pa, 1.0 MPa and 1500 Pa, respectively. When actin accumulations are involved, Young’s modulus can vary/increase over 4–5 orders of magnitude [48], which is ascribed to cells actively controlling length and density of the actin filaments network in some specific conditions [49]. The size of nanofibers was assumed from fluorescence microscopy of Appendix A and separately from the AFM investigation in Appendix A (4.0 μm length and 1.0 μm rod diameter in bundle state). We first simulated the cell body in the absence of nanofiber, producing a force-indentation curve in good agreement with that experimentally measured in Figure 4d, validating our computation. We then simulated cell body internalized with nanofiber, producing a force-indentation curve that is much softer than the experimentally measured one. Several simulations were then produced by varying the thickness of actin wrap guided by CLSM, from which we extrapolated a thickness range between 100 and 1000 nm. With 500 nm-thick actin wrapping around the nanofiber, the force-indentation curve agrees with experimental curve well (Figure 4d), and the corresponding normal stress field under a force of 3.7 nN is shown in Figure 4c. Additional simulation data can be found in Appendix A, showing that, after maintaining the indentation length at 600 nm, normal stress for a cell containing wrapped nanofiber is higher than those without actin wrap. As such, the higher Young’s modulus observed is a result of mechanical convolution of nanofiber with actin filaments recruited inside the external protrusion during the internalization process. 

As it is known that actin provides the driving force for cellular mechanical action [49], the higher density of the actin wrap may be responsible for the forces driving the internalization. In fact, once internalization is initiated, the external cell membrane is stretched and folded in order to accommodate the external body, and actin is indeed the motor used in this process. After internalization, the phagocytic load undergoes the maturation process until final fusion with lysosome for degradation. During the lifespan of a phagosome, actin is heavily involved to facilitate the movement, migration, docking and fusion with lysosomes or other organelles [46,47,50,51]. Therefore, the increase of actin could be functional for the internalization process. For example, it was evidenced that knocking down actin nucleation gene [46] or chemically depolymerizing actin (with cytochalasinD [46] or latrunculinB [51]) caused the block of phagosomes’ formation and eventually inhibited the endocytosis process. Thus, our finding verifies the importance of actin in physiological response, especially the cellular mechanical variation, experimentally and theoretically, which is induced by the long-neglected mechanical effect induced by drug carriers. The potential application of our findings in drug delivery strategies could involve actin redistribution around the drug carrier. For instance, as it is reported that the actin network provides compressive force when it covers the internalized substance [52], the nanofiber’s mechanical property can be adjusted during material synthesis to induce a long-lasting actin coat on its surface, resulting in drug release triggered by deformation and specifically focused on macrophage population.

## 4. Conclusions

We used PLGA-PEG hydrogel nanofibers as a model drug carrier to study the macrophage mechanical property after internalization. The principal findings of this study relate overall/local mechanical property variation to actin redistribution. The following observations were made: (i) increase of Young’s modulus colocalized at nanofibers, (ii) actin cytoskeleton distribution was affected by nanofibers and (iii) FEM simulation required an additional wrap layer around the nanofiber to reproduce AFM results. The interplay between nanofiber internalization and cell mechanical response revealed that cytoskeletal elements, such as actin, can be influenced by both chemical and physical means. The recruitment and redistribution of actin cytoskeleton is fundamental during this response, evidenced by a local increase of actin enwrapping the phagocytic load. Actin is a necessary element involved during formation, maturation and movement of phagosomes within cytosol. Our results also demonstrate the efficacy of mapping and nanomechanical investigation of cytoskeletal organization through a combination of AFM, CLSM and FEM as a means to understand the mechanism of internalization of nanoparticles, in particular highlighting AFM sensitivity to actin cytoskeletal reorganization.

## Figures and Tables

**Figure 1 materials-13-03235-f001:**
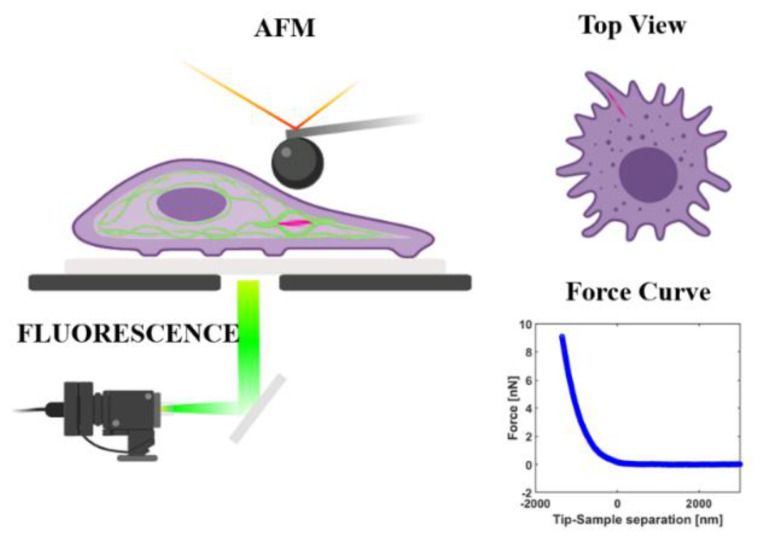
Schematic of the atomic force microscopy (AFM) experiment coupled with a fluorescence microscope. A cell containing nanofibers is visualized under the microscope, while a force curve is simultaneously acquired from AFM indentation.

**Figure 2 materials-13-03235-f002:**
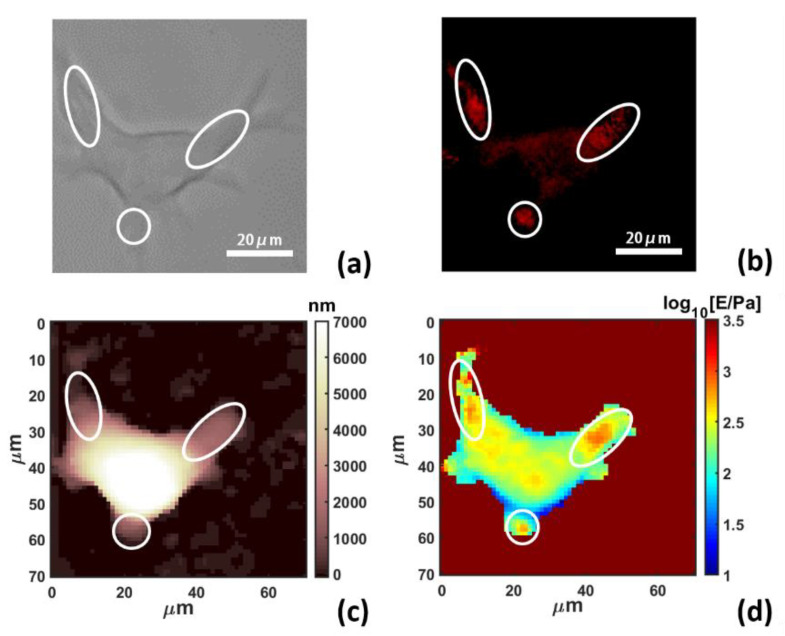
Simultaneously acquired optical microscopy images and AFM mappings of a selected macrophage after 14 h internalization of PLGA-PEG nanofibers, (**a**) transmission optical image, (**b**) fluorescence microscopy with green excitation for Nile Red, (**c**) AFM topography mapping and (**d**) Young’s Modulus mapping.

**Figure 3 materials-13-03235-f003:**
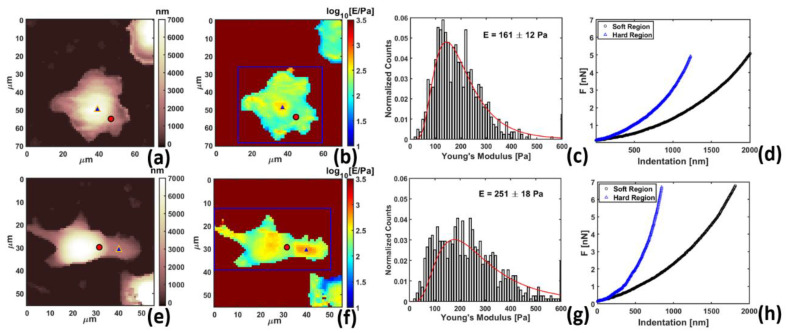
Mechanical comparison of (**a**–**d**) control cells and (**e**–**h**) cells after 14 h internalization with 100 μg/mL PLGA-PEG nanofibers, (**a**) and (**e**) zero force morphology mapping, (**b**) and (**f**) Young’s modulus mapping, (**c**) and (**g**) histogram of Young’s modulus of cells contained in the blue box in (**b**) and (**f**) with log-normal fit, and (**d**) and (**h**) force vs. indentation curves for soft region (black circles) and hard region (blue triangles).

**Figure 4 materials-13-03235-f004:**
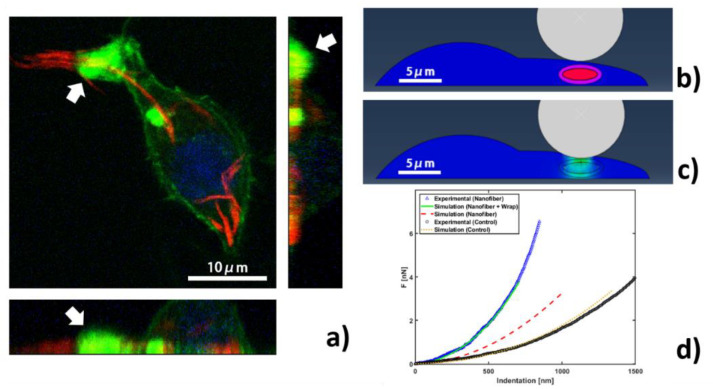
The effect of actin aggregation around nanofibers, (**a**) three-dimensional (3D) stacking confocal laser scanning microscopy (CLSM) images of fixed cells showing the internalized PLGA-PEG nanofibers and actin distribution (Scale bar: 10 μm. Blue: DAPI for nuclei, green: Phalloidin for actin, red: Nile Red for nanofibers, white arrow shows actin wrap around nanofiber). (**b**) Finite Element Method (FEM) model section showing the probe indenting cell (blue) with internalized nanofiber (red) wrapped by actin layer (purple), (**c**) normal stress field plot for FEM indentation at force 3.7 nN of nanofiber wrapped by 500 nm layer, (**d**) force vs. indentation curves acquired experimentally on top of nanofiber (blue triangles) and in cell body (black circles), in comparison with three different FEM simulations of cell body (purple dotted line), cell with nanofibers (red dashed line) and cell with nanofibers wrapped by actin (green solid line).

**Table 1 materials-13-03235-t001:** Statistical analysis of Young’s modulus for RAW264.7 with/without internalization of nanofibers.

Parameter	Control	Cell with Nanofibers
Concentration (mg/mL)	NA	100
Population	30	20
Mean value Young’s Modulus (Pa)	148.8	215.9
Standard Deviation (Pa)	23.9	35.5
Standard Error of Mean (Pa)	4.4	7.9
*p*-value (comparing to control)		<0.0001 (****)

(****) significance level <0.0001

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
