# Peer review of "Spatially Resolved Correlation between Stiffness Increase and Actin Aggregation around Nanofibers Internalized in Living Macrophages"

_materials, 2020, doi:10.3390/ma13143235_

Round 1

Reviewer 1 Report

The authors used AFM integrated with fluorescence microscope to simultaneously map the cell Young’s modulus and morphology in correspondence to the localized PLGA-PEG nanofibers. The findings are aligned with the existing literature. The manuscript reads well and has enough data for publication.

Some example of force curves on the different regions of the cells (control and modified) would help the reader to better understand the AFM data.

Line 295: please explain “fixing indentation length”

Reviewer 2 Report

The manuscript “Spatially resolved correlation between stiffness increase and actin aggregation around nanofibers internalized in living macrophages” by G. Zhou et al. describes a detailed study about the cytoskeleton reorganization of macrophages, with specific reference to actin molecules, as a consequence of biocompatible (PLGA-PEG) nanofibers internalization. The topic is relevant from a biophysical point of view considering the role of macrophages in immune system, especially in fighting pathogens and tumor development, and has been faced out both experimentally and by simulations. Confocal microscopy has been used to get an insight to nanofibers internalization and actin recruitment around nanofibers was demonstrated. Topography reconstruction of macrophages and their nanomechanical properties have been measured by AFM - coupled with fluorescent microscopy - revealing an increase of cell stiffness for sample treated with nanofibers. FD curve analysis reveals an increase in Young’s modulus whose estimation is well explained and properly referenced within Results section. Finite element simulations taking into account geometry and mechanical measurements have been performed and are in accordance with experimental observations as the presence of a wrapping layer around nanofibers is required to reply the force dependence behavior shown in AFM analysis. The manuscript is easily readable, well-written and well-organized. The experimental procedures are clear  and well-described. Figures and captions are clear and appealing. The discussion is in general well-founded even if in some aspects is a little bit too qualitative; conclusions are well-supported. So, I think that the manuscript is suitable for publication in Materials Journal after some minor revisions needed to clarify some points that are listed below:

1) In the Introduction, authors refer to AFM as a powerful biophysical technique 'to detect variation of mechanical properties of actin correlated to pathophysiological conditions...' For completeness, they should also consider works done specifically on cytoskeleton variations induced by drugs with single cell force spectroscopy approach (https://doi.org/10.1002/jmr.2173 - Journal of Molecular Recognition 2012).

2) In Results section (line 218) authors assert 'Control exhibited Young’s Modulus of 161 ± 12 Pa.... This was much softer than other cell lines'. This value is very low indicating these cell as very soft but what do the authors refer to with 'other cell lines'? Do they always refer to macrophages or other cell types? If the authors refer to macrophages, thus indicating this cell line as the softest of all macrophages, then the sentence that comes after has to be rewritten 'such low Young’s modulus may reflect the structural organization of actin in macrophages for higher deformability' since this property should be proper of all macrophages, and this does not explain why this cell line has this exceptional low stiffness value. If instead they refer to other types of cells, the concept is clear, but it must be made explicit in the text, so please rephrase.

3) Figure S1 show the correlation between internalized nanofibers and the increase of Young's modulus of the cell reported in Figure 3. Do all the panels of Fig.S1 belong to the same cell? Because panels b) and d) seem to be inconsistent; in particular, the high intensity red signal in the upper part of the cell (b) suggests an high presence of nanofibers but in the corresponding Young’s Modulus mapping (d) the same region does not show high stiffness value. Do the authors have any explanation about that?

4) In Results section (line 265) 'Such evidence in our case came from the yellow color...' for greater clarity, authors could insert a marker (e.g. an arrow) in one of the panels a) of Figure 4 indicating the area of interest.

5) In Results section (line 266) '... indicating that nanofibers can induce a redistribution of actin to internalize the nanofibers located in an external cell protrusion'. Authors should try to quantitatively analyze such actin redistribution for example using the intensity of fluorescent signals. By evaluating the difference in signal intensity between control and sample with nanofibers, an estimation of the signal increase and consequently a percentage increase due to aggregation can be obtained. This estimation would give a more quantitative notation, useful for potential comparisons with other works.

6)In Conclusion section, authors stated 'the increase of actin we measured in this study could exert internal forces during phagosome formation...'. The measurements shown in the paper reveal that actin cytoskeleton is involved in the nanofibers internalization process by wrapping them and a molecules recruitment has been shown; the increase of actin molecules around the fibers detected here cannot be related to internal forces involved in phagosome formation; there are no measurements about it. Obviously actin will be involved, but in this form it is too speculative. Please, rephrase.

Reviewer 3 Report

Dear author:

In the paper “Spatially Resolved Correlation between stiffness increase and actin aggregation around nanofibers internalized in living macrophages”, the authors are using atomic force microscopy combined with a confocal laser scanning microscopy to inspect in the mechanical properties of living macrophages using PLGA-PEG hydrogels nanofibers.

As a novelty they have found that actin changed the distribution around the PLGA-PEG and this could be an important property to develop novel strategies for drug delivery.

They have claimed that there was an increase of Young´s modulus at nanofibers because of the actin redistribution.

AFM integrated with fluorescence microscopy allows to correlate the structure change and quantitative nanomechanical variation induced by the nanofiber internalization.

Also, it is possible to look for a proportional correlation between fluorescence and stiffness. However, this is not done.

It is not clear how this discovery will improve or enhance the develop or different drug delivery strategies.

There is not discussion about similar publication or results. Figures and schemes can be improve, as well as, data representation. 

In my opinion the method and statistic seem to be correct. However,  I don’t consider myself with the enough expertise in the topic to judge the current paper and decide if this should be published or not. 

Your Sincerely, 
